# Characteristics and Dynamics of University Students' Awareness of Retired Mobile Phones in China

Ang Li [1], Bo Li [1,*] , Xia Liu [2,*], Ying Zhang [3], Haiyan Zhang [4], Xuyang Lei [1], Suxia Hou [1] and Bin Lu [5]

1   Department of Resources and Environmental Engineering, Xingtai Polytechnic College, Xingtai 054000, China
2   School of Mathematics and Information Technology, Xingtai University, Xingtai 054000, China
3   School of Resources and Environment, Xingtai University, Xingtai 054000, China
4   School of Economy and Trade, Xingtai University, Xingtai 054000, China
5   State Key Laboratory of Urban and Regional Ecology, Research Center for Eco-Environmental Sciences, Chinese Academy of Sciences, Beijing 100085, China
*   Correspondence: biobo@foxmail.com (B.L.); seamoonliu@foxmail.com (X.L.)

**Abstract:** In order to obtain accurate data about university students' awareness of retired mobile phones in China, a survey was conducted in both 2012 and 2020. There were 1011 respondents in 2012 and 1830 in 2020 that completed the questionnaire, respectively, aged 18–30 years old. This work proposes the following conclusions: lifetime is reduced from 1.95 years in 2012 to 1.92 years in 2020, with a standard deviation of 1.12 in 2012 and 0.99 in 2020; broken is the most common reason for replacements, but there are also many replacements caused by poor function, poor model, or stolen, while more than half of the retired mobile phones are stockpiled, instead of being donated or sold; voucher or cash incentives are the most efficient, while environmental and charitable incentives are also efficient; when participating in take-back services with mobile phones manufactured in the last five years, or even with broken ones, the expectant amount of cash refund is 20–300 CNY in 2012 and 20–500 CNY in 2020; most results are consistent with previous research, while the distinctions are important and helpful; policy implications are proposed to improve sustainable WEEE management systems.

**Keywords:** WEEE; e-waste; retired; phone; awareness; dynamics

## 1. Introduction

After being created by Martin Cooper in 1974, the mobile phone became the most common item of electronic equipment, especially with the rapid expansion of the mobile telecommunication industry in the 21st century. The amount of mobile phone subscribers increased from 738.60 million at the end of 2000 to 8152.00 million at the end of 2020, with the penetration of subscribers increasing from 12.04% to 104.90% in the same period [1]. As a result of the rapid development, China has had the maximal manufacturing capacity of mobile phones for several years. The amount and penetration of mobile phone subscribers increased from 84.53 million and 6.77% at the end of 2000 to 1594.07 million and 112.90% at the end of 2020, respectively [1].

The characteristics of mobile phones lead to a short lifetime and frequent replacements, contributing to a great generation of retired mobile phones, which is a typical category of small waste electrical and electronic equipment (WEEE) with characteristics such as small volume, great amount, high reuse value and low recovery rate. Retired mobile phones account for a tiny share of the total weight of WEEE, associated with a non-negligible contribution to overall environmental impacts and healthy risks, while considering the great amount of potential contaminants, such as heavy metals, around the world [2–4]. Moreover, retired mobile phones are also an indispensable part of urban mines, with a great amount of bulk, precious, rare and rare-earth metals [5]. Hence, there have been global attempts to establish sustainable management systems for retired mobile phones

with legislations and policies, such as the WEEE directive in China and the European Union [6,7].

Retired mobile phones are generated in the process of consumer consumption. Consumer behavior is the direct and primary factor affecting the consumption and retirement of mobile phones. Generational characteristics are fundamental, but difficult to obtain. In order to obtain accurate national and international data, separate surveys on retired mobile phones were conducted in several countries. Previous research revealed the relation between characteristics and sociodemographic aspects such as age, gender, income, education level, etc. However, the results were difficult to extend in the spatial and temporal scales, so separate surveys focusing on a particular country or region at a certain period were still needed to contribute to the literature.

University students are long-term users of electrical and electronic equipment, showing a high consumption due to exposure to technology. They are representative, but have been ignored in environment-related research [8–11]. Therefore, it is important to help university students understand the importance of sustainable WEEE management systems, and identify the characteristics and dynamics of university students' awareness of retired mobile phones. Therefore, an exploratory survey about awareness, knowledge and participation of retired mobile phones was conducted with university students in both 2012 and 2020, with the aim of revealing the characteristics of university students' awareness of retired mobile phones. Furthermore, the dynamics of university students' awareness were also discussed by using a comparative study between data from surveys on different temporal scales, with the aim of contributing to the literature, and promoting sustainable WEEE management systems among young consumers.

## 2. Materials and Methods

### 2.1. About the Questionnaire

For better understanding, a questionnaire in Chinese was used to collect the data. The framework and construct of the questionnaire followed principles of questionnaire design [12] and previous research [13]. The content and type of questions, which are shown in Table 1, were referenced previous research [13] for the convenience of the comparative study.

**Table 1.** The index, content and type of questions in the questionnaire.

| Index | Question Content | Question Type |
|---|---|---|
| Section 3.1 Table 2 | Essential information, such as age, gender, etc. | Multiple choices with unitary answers |
| Section 3.2 Figure 1 | The lifetime of the most recent retired mobile phones | Multiple choices with unitary answers |
| Section 3.3 Figure 2 | The frequency of mobile phones replacements | Multiple choices with unitary answers |
| Section 3.4 Figure 3 | The reasons for mobile phones replacements | Multiple choices with multiple answers |
| Section 3.5 Figure 4 | The disposal options for retired mobile phones | Multiple choices with unitary answers |
| Section 3.6 Figure 5 | The reasons for stockpiling retired mobile phones | Multiple choices with unitary answers |
| Section 3.7 Figure 6 | The effects of the incentives | Multiple choices with multiple answers |
| Section 3.8 Figure 7 | The expected cash refunds from the paid take-back services | Multiple choices with unitary answers |
| Section 3.9 Figure 8 | The awareness of sharing mechanism of recycling costs | Multiple choices with unitary answers |
| Section 3.10 Figure 9 | The proportion of sharing mechanism of recycling costs | Multiple choices with unitary answers |
| Section 3.11 Figures 10 and 11 | The awareness of other environmental affairs | Multiple choices with unitary answers |

### 2.2. Data Collected Method

The questionnaire was conducted online in 2012 and 2020, which can be found at https://www.wjx.cn/jq/1370980.aspx (accessed on 22 February 2022) and https://www.wjx.cn/jq/34114081.aspx (accessed on 22 February 2022), respectively. The questionnaire

was accessible to respondents all across China belonging to high-rank educational institutions; 1011 and 1830 valid responses were collected, respectively. With the help of a detection mechanism on the logicality and integrity of the questionnaire collected from the website, all the responses were available and used in this work.

## 3. Results and Discussions

### 3.1. The Respondents' Demographic Information

The aim of the questionnaire was to assess the awareness of the use and disposal of mobile phones among university students in China, so all the respondents were 18–30 years old. The education level of the respondents could be classified as bachelor's, master's, Ph.D. and other, while the discipline of the respondents could be classified as social sciences, natural sciences and environment-related. The detailed information of respondents is showed in Table 2; the sample sizes in 2012 and 2020 were 1011 and 1830, respectively.

**Table 2.** Detailed information of respondents.

| Group | Details | 2012 | Percentage | 2020 | Percentage |
|---|---|---|---|---|---|
| Gender | Male | 668 | 66.07% | 1197 | 65.41% |
| | Female | 343 | 33.93% | 633 | 34.59% |
| Age | 18–24 | 653 | 64.59% | 1236 | 67.54% |
| | 25–30 | 358 | 35.41% | 594 | 32.46% |
| Degree | Bachelor's | 478 | 47.28% | 974 | 53.22% |
| | Master's | 430 | 42.53% | 718 | 39.23% |
| | Ph.D. | 96 | 9.50% | 123 | 6.72% |
| | Other | 7 | 0.69% | 15 | 0.83% |
| Discipline | Social sciences | 512 | 50.64% | 994 | 54.32% |
| | Natural sciences | 377 | 37.29% | 506 | 27.65% |
| | Environment-related | 122 | 12.07% | 330 | 18.03% |

### 3.2. The Lifetime of Mobile Phones

The lifetime of electrical and electronic equipment has different definitions from various viewpoints. The concept of service lifetime used in this work was found to be applicable for the generational characteristics of retired mobile phones.

The lifetime distribution of the last retired mobile phones is shown in Figure 1, in which more than one-third of retired mobile phones had a lifetime between one and two years (34.42% in 2012 and 35.46% in 2020), followed by more than one-quarter of the retired mobile phones having a lifetime between two and three years (25.89% in 2012 and 31.53% in 2020). Additionally, nearly one-sixth (18.55% in 2012 and 16.17% in 2020) and nearly one-tenth (10.52% in 2012 and 8.96% in 2020) of retired mobile phones had a lifetime between half a year and one year and three and four years, respectively. The total distribution of other lifetimes (in half a year, four to five years and five years or more) was less than one-tenth, both in 2012 and 2020.

The lifetimes of half a year, half a year to one year, one to two years, two to three years, three to four years, four to five years and five years or more were valuated to 0.25, 0.75, 1.50, 2.50, 3.50, 4.50 and 5.00, respectively, quantifying the lifetime of the retired mobile phones. The results showed that the lifetime reduced from 1.95 years in 2012 to 1.92 years in 2020, with the standard deviation of 1.12 in 2012 and 0.99 in 2020.

Compared to a previous piece of research with 440 respondents in Australia, 63.00% of the respondents mentioned that the lifetime of their phones was 2–3 years, 25.00% indicated that the lifetime was 4–5 years, while 7.00%, 4.00% and 1.00% reported lifetimes of 1 year, 6–8 years and more than 8 years, respectively. The average possession lifetime was 3.17 years (including storage), with a standard deviation of 1.50 [14], which was longer than China with the broader extension including storage in the concept of an average possession lifetime.

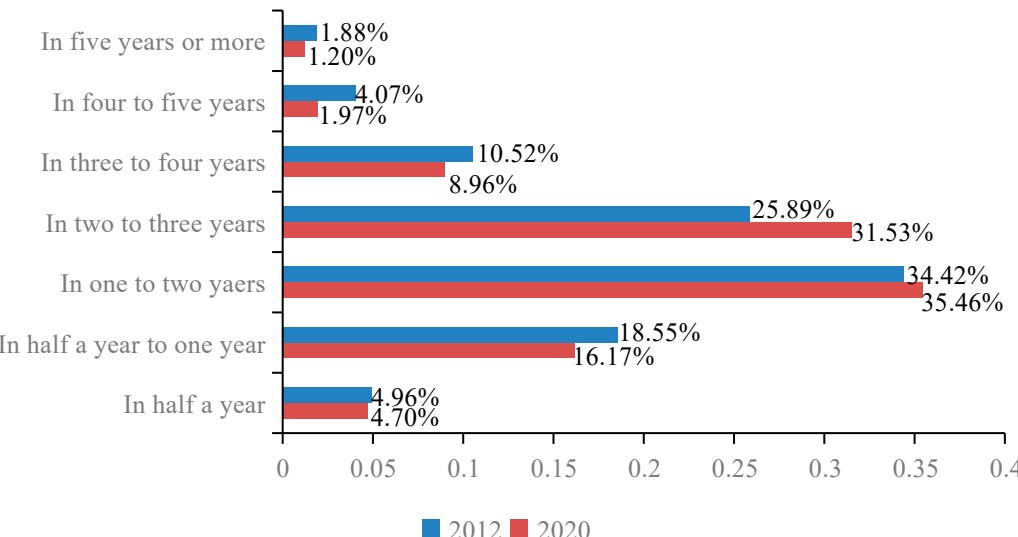

**Figure 1.** The lifetime distribution of the retired mobile phones.

The previous research showed that the typical lifetimes of mobile phones were always between 1.50 and 3.00 years, which was significantly shorter than their physical lifetime. For specific countries and regions, the lifetimes of mobile phones are always different and irregular. For example, before 2002, the lifetimes of mobile phones were 1.70 years in Japan, 3.00 years in China and 2.30 years in the U.S. [15] and west Europe [16]; then, before 2008, the lifetime of mobile phones was 1.50 years or 17.50 months in the U.S. [17]. In Asia, the lifetimes of mobile phones were 1.70 years in Japan before 2002, 2.40 years or 28 months in South Korea before 2010 [18], 1.60 years or 19 months in Indonesia before 2013 [19] and 1.90 years in China before 2015 [20]; then, in Europe, they were 3.63 years in the Czech Republic before 2012 [21], 2.70 years in Austria before 2018 [22], 16.36 months in the Netherlands before 2019 [23] and 3.00 years in Denmark before 2020 [24], while in Africa, they were 4.00 years in Nigeria before 2008 [25] and 2.30 years in Rwanda before 2016 [26].

*3.3. The Frequency of Mobile Phones Replacements*

According to the lifetime, it can be speculated that the frequency of replacements can prove the results of the lifetimes from another perspective. However, the frequency of the replacements of retired mobile phones is difficult to quantify to obtain the results of the lifetimes.

The frequency of the replacements of retired mobile phones is showed in Figure 2. The results showed that replacements occurred twice or more per year (6.17% and 6.72%) and every one to two years (26.01%) in 2020, compared with 5.84%, 3.36% and 21.86% in 2012, respectively. More than one-third (37.69% in 2012 and 34.48% in 2020) of the replacements occurred within two to three years, while more than one-fifth (21.86% in 2012 and 26.01% in 2020) of the replacements occurred within one to two years. There were less replacements once per year (10.49%) and every three years or more (16.12%) in 2020. More than three-fifths (59.55% in 2012 and 60.49% in 2020) of replacements occurred within one to three years, which could verify the lifetime of retired mobile phones from another point of view.

Compared to previous research with 2181 and 498 university student respondents in the U.K. and Germany, the results showed there were more replacements in one year in the U.K. (28.00% for once per year, 4.00% for twice per year, 1.00% for thrice per year and 1.00% for more than thrice per year) [13] and China than in Germany (3.80%) [27]. The replacements that occurred more than once a year were 76.61% in 2020 and 78.14% in 2012, compared to 66.00% in the U.K. [13] and 96.18% in Germany (45.60% for one to two

years, 30.50% for two to three years, 11.65% for three to four years and 8.43% for more than four years) [27], which showed the distinction between countries.

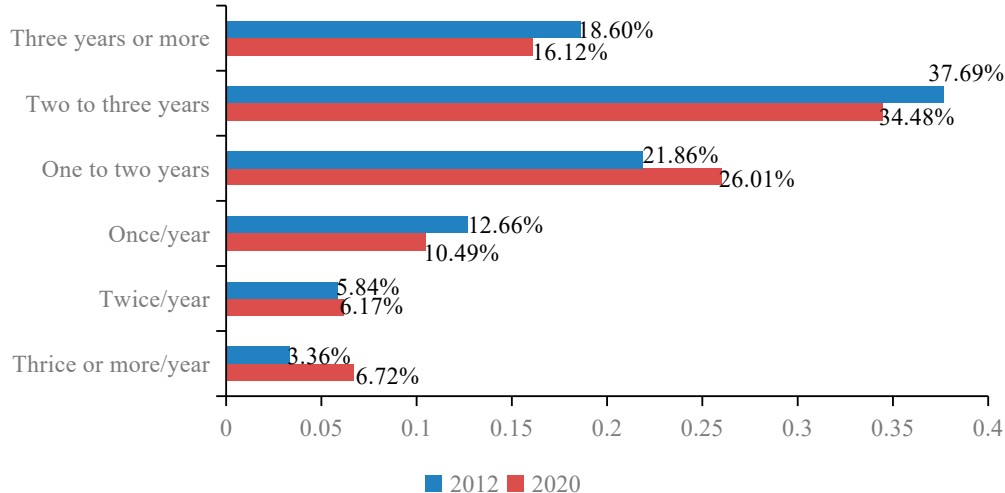

**Figure 2.** The frequency of mobile phone replacements.

*3.4. The Reasons for Mobile Phones Replacements*

Figure 3 shows the reasons for mobile phones replacements, where the most common reason was the replacements of broken phones, accounting for 65.58% in 2012 and 54.32% in 2020, as well as 30.74%, 90.42% and 57.7% in Australia [14], India [28] and the U.K. [13] respectively. Broken was the most common reason leading to replacements, which was almost unaffected by the discrepancies in temporal, spatial and other factors. By combining this point of view with the results in Figure 2, it could be deduced that many mobile phones used by university students were broken in three years or more. Although this phenomenon decreased in 2020, it still took place more than half of the time, and remained the most common reason. Reasons causing the high damage rate for mobile phones in a relatively short lifetime should be explored in future research.

Replacements caused by poor function increased from 41.44% in 2012 to 52.46% in 2020, which corresponded with the reality that respondents pursue better function with progress in technology in Australia [14], India [28] and the U.K. [13].

Replacements caused due to lost decreased from 37.29% in 2012 to 26.28% in 2020, sharing the same trend with broken. However, there were few replacements caused due to lost phones in Australia [14], Brazil [29] and the Netherlands [23] and the U.K. [13].

Replacements caused by contract decreased from 10.58% in 2012 to 3.06% in 2020, while replacements caused by network upgrade increased from 2.57% in 2012 to 12.24% in 2020, both according to the reality of the development of the telecommunication industry in China, because the majority of mobile phones in China are bought by consumers instead of provided by operators with contract in developed countries.

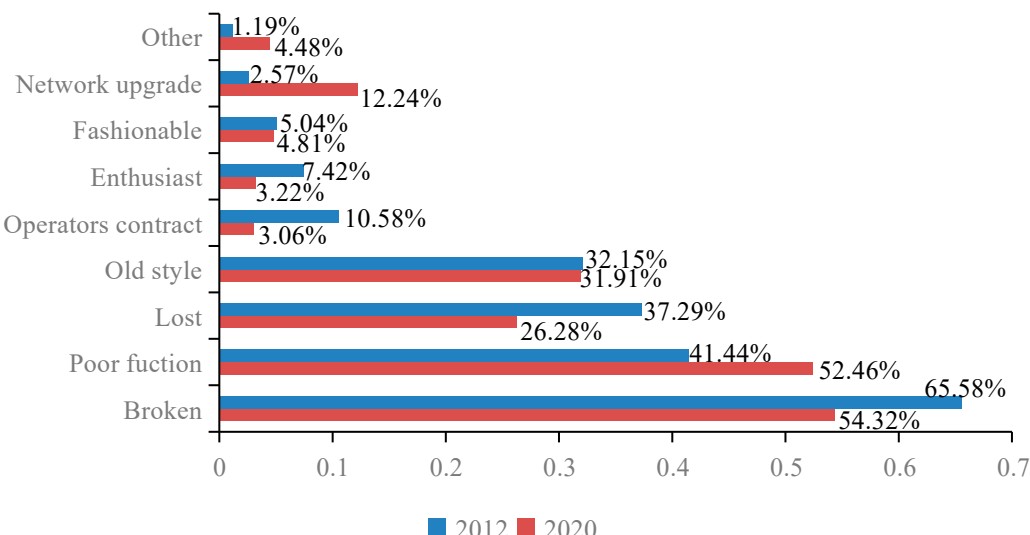

**Figure 3.** The reasons for mobile phones replacements.

*3.5. The Disposal Options for Retired Mobile Phones*

Figure 4 shows the disposal options for retired mobile phones, where more than half of retired mobile phones were stockpiled, consistent with global conditions. The proportion of stockpiled phones decreased from 64.00% in 2012 to 52.13% in 2020, with the improvement in other options. However, this proportion was relative high also in Australia [14], Brazil [29], Finland [30], Germany [27], India [28], the Netherlands [23], the U.K. [13] and the U.S. [15].

Donations or given away to individuals or organizations gratis were the second common disposal options for retired mobile phones in China, with the proportion increasing from 24.63% in 2012 to 27.16% in 2020. This phenomenon was also consistent with conditions in Brazil [29], Finland [30], Germany [27], India [28], the Netherlands [23], the U.K. [13] and the U.S. [15], while being relatively low in Australia (3.94%) [14].

The proportion of phones both sold for money or for a discount increased with the development and popularization of the internet, being 8.96% and 6.72% in 2020 compared with 7.32% and 0.89% in 2012. The proportion of sold for money phones was also consistent with conditions in Australia [14], Brazil [29], Finland [30], the Netherlands [23], the U.K. [13] and the U.S. [15], while being relatively high in Germany (33.30%) [27] and India (29.94%) [28]. The disposal option of selling a phone for a discount seemed to be seldom used, only being found in Australia (1.20%) [14] and the U.K. (5.20%) [13].

The amount of retired mobile phones sold or sent for safe disposal was quite low for both, accounting for 2.62% in 2020 and 4.75% in 2012. This phenomenon was also consistent with conditions in Brazil [29], Finland [30], Germany [27], India [28], the Netherlands [23], the U.K. [13] and the U.S. [15]. However, 42.20% and 25.70% of retired phones were sent for safe disposal in Finland [30] and Australia [14], showing that there may be some helpful experiences therein.

Notably, the results showed that 1.31% of retired mobile phones were still thrown away in general waste, compared with 2.77% in 2012. Usually, only a few retired phones were thrown away in general waste [31], but due to their large amount and complex composition, there are still potential risks associated to environment quality and human health, which may impose heavy burdens on general waste management systems, especially when there were 21.52% and 11.34% retired phones being thrown away in general waste in Australia [14] and the U.S. [15].

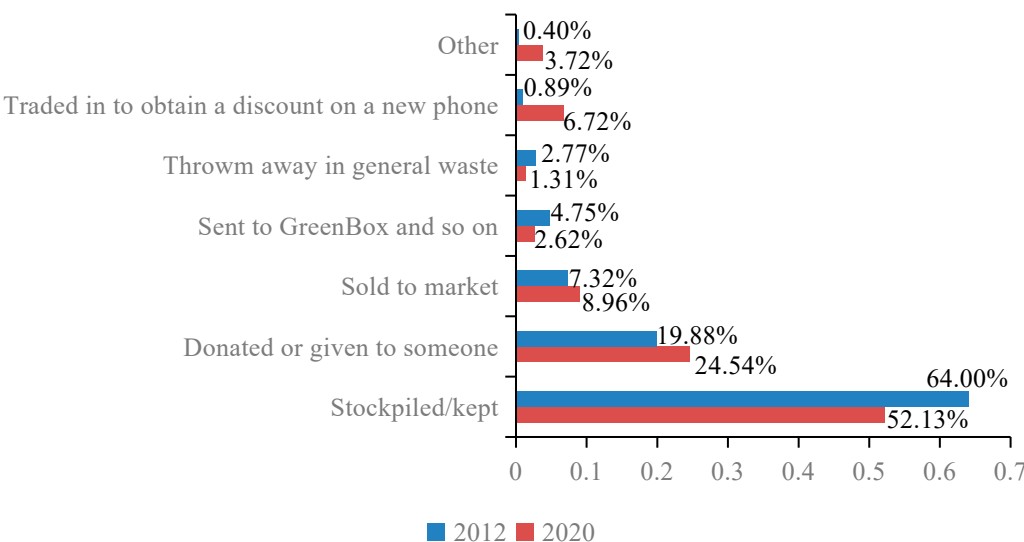

**Figure 4.** The disposal options for retired mobile phones.

*3.6. The Reasons for Stockpiling Retired Mobile Phones*

Reasons for stockpiling retired mobile phones are summarized in Figure 5, showing that the most common reason was "don't know how to deal with their retired mobile phones", both in 2020 (34.66%) and 2012 (59.81%). Compared with previous research, this phenomenon was also common in Finland [30] and the U.K. [13], while seeming to influence fewer people in Austria [22] and the Netherlands [23].

Stockpiling "for alternative use" use was also common (28.13% in 2012 and 22.20% in 2020) in China, while seeming to influence more people in Austria [22], Finland [30], the Netherlands [23] and the U.K. [13].

Stockpiling caused by "take-back services being inconvenience" decreased to 17.59% in 2020 from 35.24% in 2012, while the stockpiling reason of "take-back services being uninviting" decreased to 6.70% in 2020 from 13.45% in 2012, both conforming with the reality that the convenience and inviting nature of take-back services improved due to the internet. However, those reasons were seldom in other countries.

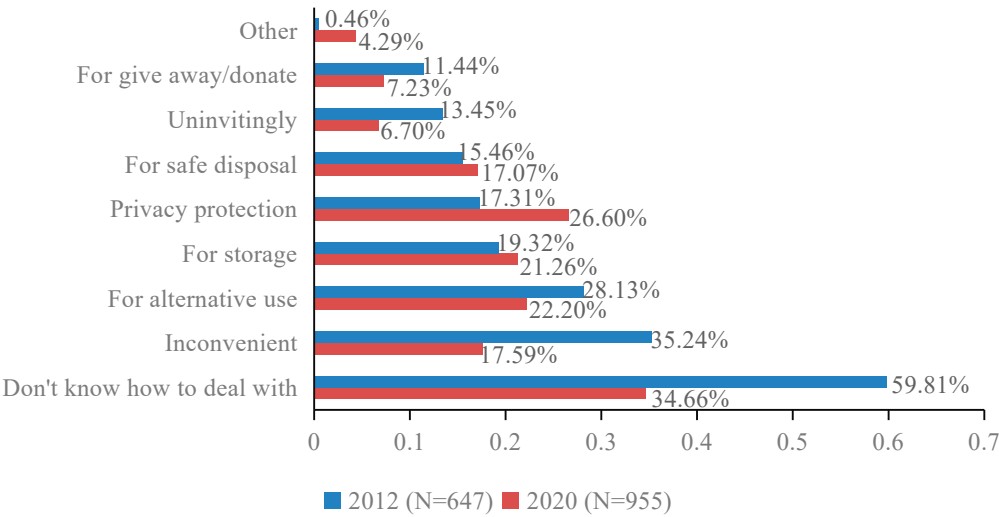

**Figure 5.** The reasons for stockpiling retired mobile phones.

On the other hand, stockpiling caused by "storage", "privacy protection" and "safe disposal" was more common in 2020, also being common reasons globally, showing the enhancement of respondents' awareness of environmental and privacy protection, and the improvement in the convenience of take-back services. The frequency and improvement in

stockpiling caused by "privacy protection" showed the global concern for cyber-security risks related to personal data, making take-back services more efficient.

### 3.7. The Effects of the Incentives

The effects of the incentives are summarized in Figure 6. The cash or voucher incentives were the most efficient, though they decreased from 2012 (81.21%) to 2020 (63.28%), which was consistent with conditions in the Netherlands [23] and the U.K. [13]. Environmental or charitable incentives were also efficient, but found to be less important than cash or voucher incentives.

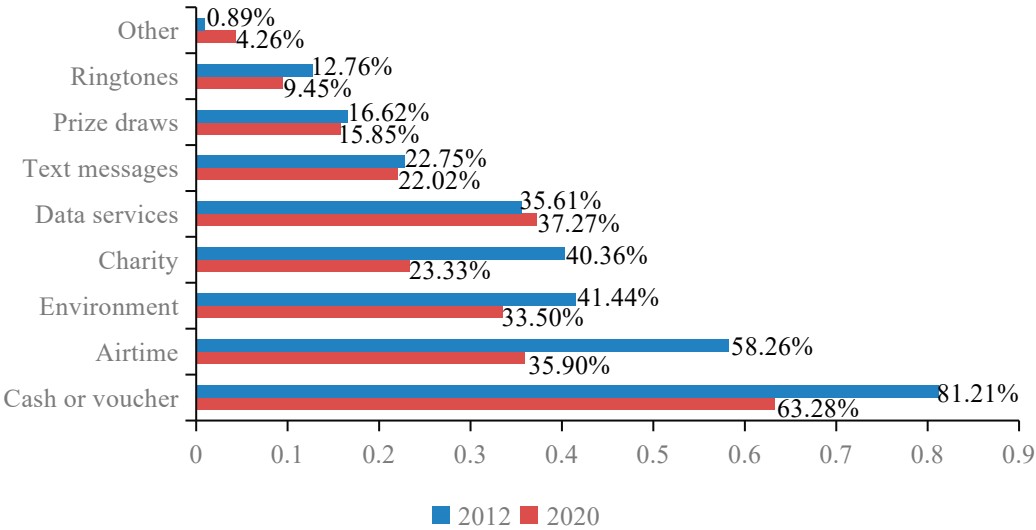

**Figure 6.** The effects of the incentives.

The effects of other incentives, including free or discounted airtime (35.90%), text messages (22.02%), prize draws (15.85%), ringtones (9.45%) and environmental (33.50%) and charitable (23.33%) incentives, all decreased in 2020, while data service incentives were more efficient in 2020 (37.27%) compared with 35.61% in 2012. The attraction of free or discounted text messages, prize draws and ringtones incentives may have not been enough for respondents in both 2012 and 2020, while the attraction of free or discounted airtime was less attractive in 2020. Pricing packages in China, including a complimentary number of free text messages and airtime, led to this phenomenon, because the pricing packages met the customers' demands. This phenomenon was in disagreement with previous research conducted in China, but was in agreement with previous research conducted in the U.K. [13]. On the other hand, both the price of and demand for text messages and airtime decreased with the competing rates offered by network operators and progress in technology. The emergence and prosperity of instant messaging apps, such as WeChat and WhatsApp, reduced the demand for airtime and promoted the demand for data services, which was also promoted by data services incentives [20].

### 3.8. The Expected Cash Refunds from Paid Take-Back Services

The expected cash refunds from paid take-back services is summarized in Figure 7. When participating in paid take-back services with mobile phones manufactured five or more years ago, or even with broken ones, 30.05% of the respondents expected to receive a cash refund of 50–100 CNY in 2020, compared with 42.24% in 2012.

For other amount offered by take-back services, such as 0–20 and 20–50 CNY, 5.46% and 15.68% of the respondents thought they were acceptable, compare with 7.52% and 30.17% in 2012, respectively. In addition, 28.80%, 16.17% and 3.83% of respondents accepted amount of 100–300, 300–500 CNY and more in 2020, compared with 13.95%, 3.56% and 2.57% in 2012, respectively.

The expected amount concentrated on 20–100 CNY in 2012 (72.41%), compared with 45.73% respondents expecting that amount in 2020. There were more expected amount concentrated on 100–500 CNY in 2020 (44.97%), comparing with 21.52% respondents expecting that amount in 2012. In previous research, 74.00% of respondents valued their mobile phone at more than 20 EUR in the Netherlands [23]. The expectant amount increased with the increasing prices of new mobile phones in both China and the Netherlands [23].

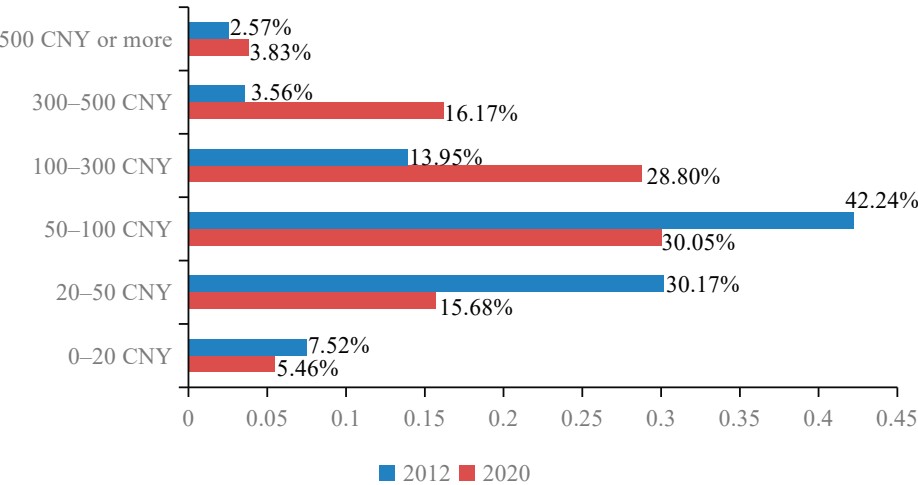

**Figure 7.** The expected cash refunds from paid take-back services.

The nominal exchange rates of CNY against USD and EUR were both approximately 1:0.15 in August 2022.

### 3.9. The Awareness of Sharing Mechanism of Recycling Costs

The awareness of sharing mechanism of recycling costs, which was chosen to represent the economic responsibility of retired mobile phones, is shown in Figure 8. Most respondents believed that the responsibility should be shared by all stakeholders, which accounted for 40.71% in 2020 and 36.70% in 2012, followed by 25.90% in 2020 and 33.53% in 2012 deemed that producers should undertake all the responsibility. The government, retailer and operators were also considered to be required to take on all the responsibility alone, accounting for 13.39%, 12.46% and 4.54% in 2020, compared with 17.90%, 6.13% and 5.34% in 2012, respectively. Only 55 respondents, accounting for 3.01%, considered that customers should take on all the responsibility alone in 2020, compared with only 0.40% in 2012, which showed the improvement in awareness. The downtrend in the awareness of governmental responsibility also showed an improvement in society, which could reduce the unnecessary responsibility placed on the government. The downtrend in the awareness of producer responsibility seemed to contradict the flourish seen in extended producer responsibility, but showed the inconvenience of producer take-back services. On the other hand, the uptrend in the awareness of retailer responsibility showed the preponderance of the take-back services offered by retailer, consistent with the reality in China.

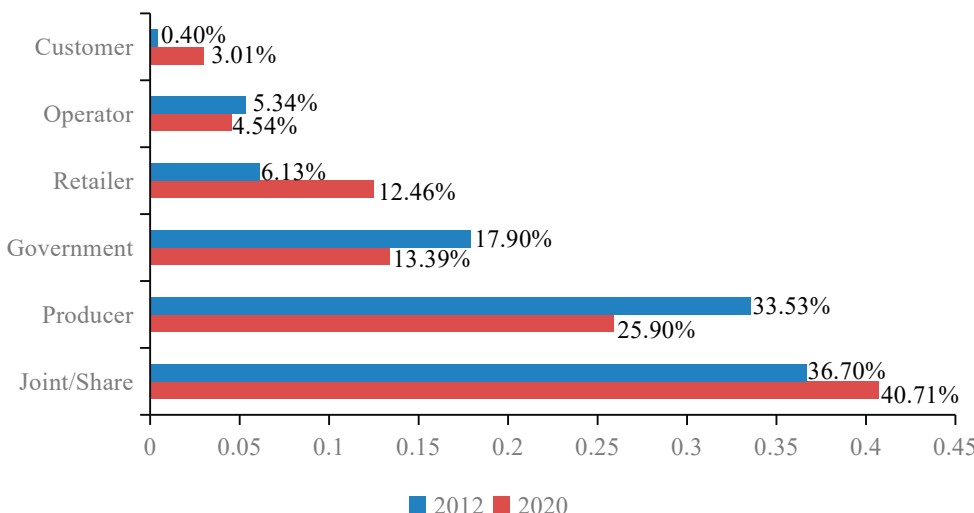

**Figure 8.** The awareness of sharing mechanism of recycling costs.

### 3.10. The Proportion of Sharing Mechanism of Recycling Costs

The proportion that customers were willing to undertake in the sharing mechanism of economic responsibility is showed in Figure 9; 46.83% of respondents in 2020 and 57.96% in 2012 could not afford more than 5% of the economic responsibility of retired mobile phones. There was an evident trend of the willingness to pay, showing a decrease as responsibility increased.

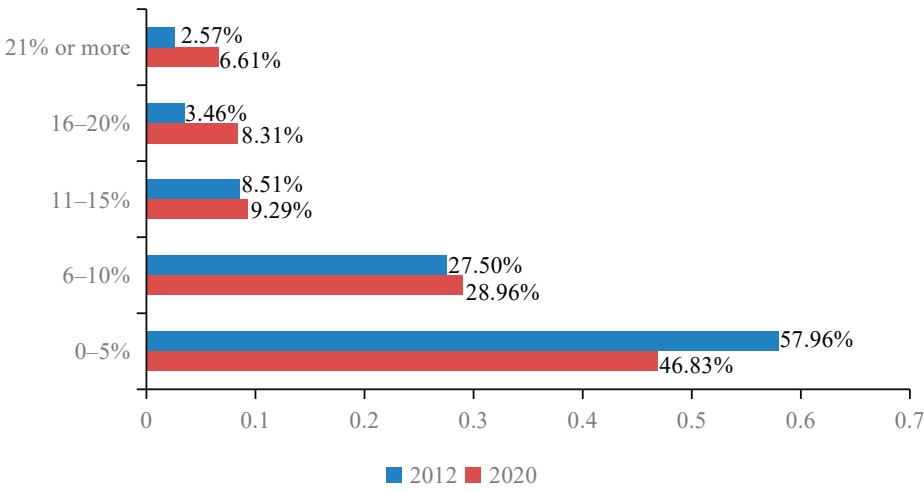

**Figure 9.** The proportion of sharing mechanism of recycling costs.

### 3.11. The Awareness of Other Environmental Affairs

The awareness of several other environmental affairs was also listed in the questionnaires, and the relevant results are shown in Figures 10 and 11. The awareness of the pollution potential of retired mobile phones remained high in 2012 (78.37%) and 2020 (79.62%), as well as the awareness of the recycling potential of retired mobile phones (59.52% in 2012 and 72.46% in 2020) and the willingness to pay for environmental quality (73.61% in 2012 and 77.27% in 2020). The relatively low awareness of the recycling potential could be due to more education being needed on environmental protection and pollution prevention than urban mining and the circular economy as common knowledge in universities [32], which met the results in Portugal [33].

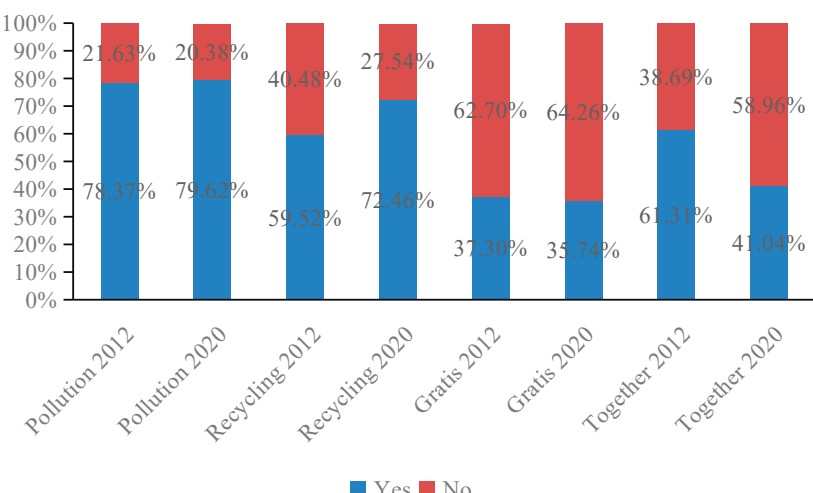

**Figure 10.** The awareness of pollution potential, recycling potential, gratis take-back services and take-back accessories together with retired mobile phones.

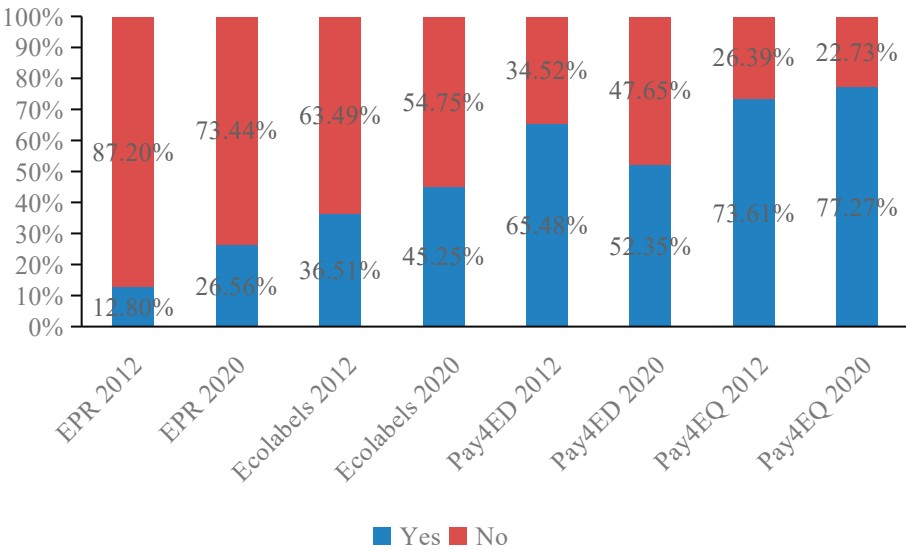

**Figure 11.** The awareness of ecolabels, extended producer responsibility, paying for environmental quality and production with ecodesign.

The awareness of the pollution potential of retired mobile phones in both 2012 and 2020 was higher than India in 2019 (64.07%) [28], which showed the effect of publicity and education in China, as well as the common concerns in both China and India as developing countries.

The awareness of extended producer responsibility and ecolabels was relatively low, and increased from 2012 (12.80% and 36.51%) to 2020 (26.56% and 45.25%).

The relatively low and decreasing (37.30% in 2012 and 35.74% in 2020) awareness of gratis take-back services of retired mobile phones correlated with the preference for cash or voucher incentives, even the low awareness of sharing mechanism of recycling costs.

The awareness of take-back accessories together with retired mobile phones decreased from 2012 (61.31%) to 2020 (41.04%), which could be due to the improvement in the convenience of take-back services.

The willingness to pay for production with ecodesign decreased from 2012 (65.48%) to 2020 (52.35%), which could be due to the improvement in the environmental performance of common production.

## 4. Conclusions

In order to obtain accurate data of university students' awareness of retired mobile phones in China, an exploratory survey was conducted in both 2012 and 2020. In total, 1011 and 1830 18–30-year-old respondents in 2012 and 2020, respectively, completed the questionnaire online. The questionnaire consisted of multiple choices with unitary or multiple answers.

This work proposed the following conclusions:

The lifetime distribution of the last retired mobile phones showed that more than one-third of retired mobile phones had a lifetime between one and two years (34.42% in 2012 and 35.46% in 2020), followed by more than one-quarter of retired mobile phones having a lifetime between two and three years (25.89% in 2012 and 31.53% in 2020). Additionally, nearly one-sixth (18.55% in 2012 and 16.17% in 2020) and nearly one-tenth (10.52% in 2012 and 8.96% in 2020) of retired mobile phones had a lifetime between half a year and one year and three and four years, respectively. The total distribution of other lifetimes (half a year, four to five years and five years or more) was less than one-tenth in both 2012 and 2020. The lifetime of retired mobile phones reduced from 1.95 years in 2012 to 1.92 years in 2020, with a standard deviation of 1.12 in 2012 and 0.99 in 2020.

According to the essence of lifetime, it could be speculated that the frequency of replacements could prove the lifetime results from another perspective. However, the frequency of the replacements of retired mobile phones was difficult to quantify to obtain the results of the lifetime. The results showed that more replacements occurred twice or more times per year (6.17% and 6.72%) and every one to two years (26.01%) in 2020, compared with 5.84%, 3.36% and 21.86% in 2012, respectively. More than one-third (37.69% in 2012 and 34.48% in 2020) of replacements occurred within two to three years, while more than one-fifth (21.86% in 2012 and 26.01% in 2020) of the replacements occurred within one to two years. There were less replacements once per year (10.49%) or every three years and more (16.12%) in 2020. More than three-fifths (59.55% in 2012 and 60.49% in 2020) of replacements occurred within one to three years, which could verify the lifetime of retired mobile phones from another point of view.

Broken was the most common reason leading to replacements, which was almost unaffected by the discrepancies in temporal, spatial and other factors. There were also many replacements caused due to poor function, poor model or stolen, while more than half of the retired mobile phones were stockpiled, instead of being donated or sold.

Many respondents stockpiled retired mobile phones for reasons such as "for alternative use", while stockpiling reasons of "take-back services being inconvenience" and "take-back services being uninviting" both decreased in 2020, according to the reality that improvements in the convenience and inviting nature of take-back services were due to the internet. On the other hand, stockpiling caused by "storage", "privacy protection" and "safe disposal" was more common in 2020, also being common reasons globally, showing the enhancement of respondents' awareness of environmental and privacy protection and improvements in the convenience of take-back services.

Cash or voucher incentives were the most efficient, followed by environmental and charitable incentives. The attraction of free or discounted text messages, prize draws and ringtones incentives may not have been enough for respondents in both 2012 and 2020, while the free or discounted airtime incentive was less attractive in 2020.

The awareness of other environmental affairs, such as ecolabels, recycling potential and extended producer responsibility, improved in 2020 due to education on environmental protection and pollution prevention. The awareness of gratis take-back services for retired mobile phones correlated with the preference for cash or voucher incentives, even the low awareness of sharing mechanism of recycling costs. The awareness of take-back accessories together with retired mobile phones decreased from 2012 (61.31%) to 2020 (41.04%), which could be due to the improved convenience of take-back services.

Most results in this work were consistent with conditions found in previous research, such as common reasons and disposal options for retired mobile phones, which showed the

common demand for sustainable WEEE management systems globally. On the other hand, the distinctions in reasons for stockpiling and effects of the incentives could be important and helpful in improving the sustainability of WEEE management systems in spatial and temporal scales.

The characteristics and dynamics of university students' awareness on retired mobile phones were identified in this work; however, there were several limitations, such as the deficiency in correlation analyses between the characteristics and essential information (age, gender, financial status, etc.). The reasons causing the high damage rate for mobile phones in a relatively short lifetime should also be explored in future research.

Previous research on the characteristics of awareness focused on national and regional scales, and the comparative study based on previous research showed both diversity and distinctions on the temporal and spatial scales. Hence, research relevant to different products and on temporal and spatial scales is still necessary in future research.

Furthermore, policy implications were proposed to improve the sustainability of retired mobile phones management systems in China. From the governmental perspective, the publicity and education on environmental affairs were efficient and should be persistent; from the industrial perspective, cash or voucher incentives were the most efficient options, which could help take-back services.

**Author Contributions:** A.L., B.L. (Bo Li) and X.L. (Xia Liu) proposed the idea; A.L., B.L. (Bo Li), X.L. (Xia Liu), Y.Z., H.Z., S.H. and B.L. (Bin Lu) collected the data; A.L. and B.L. (Bo Li) wrote the manuscript; X.L. (Xia Liu), Y.Z., H.Z., X.L. (Xuyang Lei), S.H. and B.L. (Bin Lu) polished the manuscript. All authors have read and agreed to the published version of the manuscript.

**Funding:** This research was funded by the National Key R&D Program of China (no. 2018YFC1903604-01 and 2019YFC1906101), the S&T Program of Hebei (no. 21553301D and 22327312D), the S&T Program of Xingtai (no. 2020ZZ047, 2022ZZ094 and XTSKFZ2022161) and the Xingtai Polytechnic College (no. 3020503 and 202202).

**Institutional Review Board Statement:** Not applicable.

**Informed Consent Statement:** Not applicable.

**Data Availability Statement:** Not applicable.

**Acknowledgments:** The authors are grateful to the honored editors and referees for their hard work and selfless help. All the anonymous respondents who responded to the questionnaire are also sincerely appreciated.

**Conflicts of Interest:** The authors declare no conflict of interest.

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
