# Peer review of "Characteristics and Dynamics of University Students’ Awareness of Retired Mobile Phones in China"

_sustainability, doi:10.3390/su141710587_

Round 1

Reviewer 1 Report

Comments are provided in the attached file.

Author Response

Dear reviewer:

We appreciate all the comments, which are valuable and helpful for improving this manuscript. We have studied all the comments carefully and have made revision with track changes in the revised manuscript.

The details of the response are in the pdf file.

Reviewer 2 Report

General comments

The research object is meaningful, and this paper investigates college students' awareness of treating retired mobile phones through questionnaires. But there are some suggestions for the paper.

Specific comments

·         In introduction part (line 41-46), the importance of studying the characteristics of retired mobile phones should be emphasized, such as listing the precious metal in retired mobile phones, and the harmful effect of retired mobile phones to the environment without proper treatment.

·         In literature part (line 48-49), the current legislation and policies to treat retired mobile phones should be specified. And the result of the paper should be helpful for policymaker, and the paper could list some suggestions to current legislation education and policies (from the view of treatment facility allocation, education level, subsidy to treatment facility), based on characteristics (eg. consumption behavior, recycling behavior) of university students’ awareness on retried mobile phones

·         For questionnaire (line 75-78), only type of the questions is explained, and the validation method (how to distinguish collected questionnaire data is valid for statistics) should be explained as well.

·         For the whole results part, there exists 11 subtitles, and flow chat for discussion is suggested to demonstrate them clearly and make readers to understand framework of discussion easily.

·         For result 3.2 (Line 97-110), the explanation of different categories of lifetime should be explained in introduction part, no need to include them in results.

·         For result 3.2 (line 114-127), results of Australia study are listed without comparison with the current study.

·         For result 3.3 (line 162-168), results are compared without summary, this comparison seems that the paper only list the previous results.

·         For result 3.9 (line 311-317), awareness results are compared with previous study without difference explanation.

·         For conclusion part (line 369-432), the paragraphs could be shortened with list the important information. For example, the sentence (line 383-386) could not be included.

Minor comments:

1.      In line 35-40, could include more references

2.      In line 48, “it’s pity that” is not scientific speaking for research gap.

3.      The value in Figure 1 is not clear, less important values can be ignored

4.      The value in Figure 2 should be arranged properly (such as 3.36%)

5.      In line 177-180, the sentence should not be in the form of question

6.      Some values in Figure 3 are not clear, the colour of pattern can be changed (such as 3.72%)

7.      For result 3.7 (line 287-289), reference can be included

8.      The value in Figure 7 should be arranged properly (such as 3.56%)

9.      The value in Figure 8 should be arranged properly (such as location of “5.34%”, “6.13%”)

10.  The value in Figure 9 should be arranged properly (such as location of “2.57%”, “3.46%”, “8.51%”)

Author Response

(The authors gave the same response as above.)

Reviewer 3 Report

The current manuscript entitled “Characteristics and dynamic of university students’ awareness on retried mobile phones in China” by Li et al. provide useful information regarding an exploratory survey about awareness, knowledge, and participation on retired mobile phones among university students in China. After a careful review, I found this work interesting. However, I found some grey points in the manuscripts which should be rectified in a revised version. My specific comments are:

·        The abstract should be rewritten. I can see the first sentence needs major improvement in terms of syntax and voice change.

·        What are the possible hazards of using outdated mobile phones? The author needs to elaborate on it in the introduction part.

·        The material and method section should be revised extensively. Provide the information about the collection of other variables with proper references. Better to provide a flowchart indicating the whole methodological process.

·        Table 1: Multiple choices with a single answer? Does the author mean by selecting only one option from the given list?

·        I think one of the most important factors is the financial status of the students which is missing from this study. Students’ behavior to use/replace/disposing of old mobile phones is more likely affected by financial stability.

·        Figure 1 and alike need to be redrawn; values are not clearly shown. I suggest it change to a Treemap chart for better visualization.

·        Why broken mobile phone is the topmost reason for their replacement? Do students use their phones during hard physical activities like sports/gaming?

·        Reduce the conclusion to not more than 300 words. Provide major findings only and future recommendations. A major part of the conclusion can be a shift to a new heading “Discussion”. Also, the environmental implications of waste generated from mobile disposal must be discussed with supporting references.

·        A few parts from the Results section where authors have compared the data with other studies need to be shifted under the new “Discussion” section.

·        References are okay.

Author Response

(The authors gave the same response as above.)

Reviewer 4 Report

Dear authors,

the topic of the submitted paper meets the focus of the journal Sustainability and is very actual. However, changes are required to meet the requirements.

POINT1 First, the abstract needs to be significantly revised. For better comprehensibility, divide it into shorter sentences and state clearly the goals focus of the study, and its limits and not just state the results.

POINTS 2 - Visualisation

Table 2. Detailed information of respondents - I would recommend adding relative frequencies for comparing the sample in the years 2012 and 2020

Figure 1. The lifetime distribution of the retired mobile phones

The figures in the graph are not readable, and this type of pie chart might confuse the reader. From a statistics point of view, it gives more information bar chart and is more precise.

Figure 2. The frequency of mobile phones replacement - the same issue.

Figure 3. The reasons for mobile phones replacement. - modify the graph so the bars will be increased/decreased to read it  better (like fig.5 or fig.6)

Figure 4. The dispose options for the retired mobile phones -not readable figures

Figure 5. The reasons for stockpiled the mobile phones replacement. -it is OK. it is your data? if not it is compulsory to mention sorry and that you are the only author of the graphical part

Figure 7. The expected cash refund from the paid mobile phone take-back services, is not readable numbers, could you add in text sum in USD/ EUR? or at least add the exchange rate or percentage of the price? It would be helpful for readers of an international journal to understand the presented results more in detail.

Fig 10 and Fig 11 don´t have outer rings as it is in their title. Figure 10. The frequency of mobile phones replacement (with data of 2020 on the outer ring).

POINT 3 The paper brings a description of results, is missing the more details research or analysis, comparison of variability or identification of some differences regarding age/gender/..

POINT 4 My suggestion on how to improve it is necessary first to propose a research question or some hypothesis and then prove it. 

POINT 4 In conclusion, are no mentioned limitations of study or further research.

Best regards

Reviewe r

Author Response

(The authors gave the same response as above.)

Round 2

Reviewer 2 Report

The paper has been revised and is fit for publication

Author Response

Dear reviewer:

We appreciate all your comments, which are valuable and helpful for improving this manuscript, thank you very much!

Reviewer 3 Report

I can see that authors have made necessary changes in the revised version of this manuscript. However, they gave non-satisfactory rebuttals for some of my comments which could be implemented for improving the quality of the work. But that do not affect the overall judgment for accepting this manuscript. I suggest acceptance on current form.

Author Response

(The authors gave the same response as above.)
